# “FastCheck^FLI^ PPR-like”—A Molecular Tool for the Fast Genome Detection of PPRV and Differential Diagnostic Pathogens

**DOI:** 10.3390/v12111227

**Published:** 2020-10-29

**Authors:** Sabrina Halecker, Thomas C. Mettenleiter, Martin Beer, Bernd Hoffmann

**Affiliations:** 1Institute of Diagnostic Virology, Friedrich-Loeffler-Institut, Südufer 10, 17493 Greifswald-Insel Riems, Germany; sabrina.halecker@fli.de (S.H.); martin.beer@fli.de (M.B.); 2Friedrich-Loeffler-Institut, Südufer 10, 17493 Greifswald-Insel Riems, Germany; ThomasC.Mettenleiter@fli.de

**Keywords:** peste des petits ruminants virus (PPRV), *Small ruminant morbilli virus*, molecular pen-side test, fast extraction, high-speed RT-qPCR, rapid detection method, differential diagnosis

## Abstract

To assist the global eradication of peste des petits ruminants virus (PPRV), a molecular test for the rapid and reliable detection of PPRV was developed which additionally enables the detection of pathogens relevant for differential diagnostics. For this purpose, the necessary time frame of a magnetic bead-based nucleic acid extraction protocol was markedly shortened to 7 min and 13 s. The optimized extraction was run on a BioSprint 15 platform. Furthermore, a high-speed multi-well RT-qPCR for the genome detection of PPRV and additional important pathogens such as Foot-and-mouth disease virus, Parapoxvirus ovis, Goatpox virus, and *Mycoplasma capricolum subsp. capripneumoniae* was established and combined with suitable internal control assays. The here-described qPCR is based on a lyophilized master mix and takes only around 30 to 40 min. Several qPCR cyclers were evaluated regarding their suitability for fast-cycling approaches and for their diagnostic performance in a high-speed RT-qPCR. The final evaluation was conducted on the BioRad CFX96 and also on a portable Liberty16 qPCR cycler. The new molecular test designated as “FastCheck^FLI^ PPR-like”, which is based on rapid nucleic acid extraction and high-speed RT-qPCR, delivered reliable results in less than one hour, allowing its use also in a pen-side scenario.

## 1. Introduction

Peste des petits ruminants (PPR) is a viral disease of domestic and wild ruminants with a wide range of susceptible species of the order *Artiodactyla* [1,2,3,4]. The causative agent is peste des petits ruminants virus (PPRV), a member of the genus *Morbillivirus* in the *Paramyxoviridae* family [5]. The genome of PPRV consists of 15,948 nucleotides, which encode for six structural (nucleocapsid protein (N), phosphoprotein (P), matrix protein (M), fusion protein (F), hemagglutinin protein (H), large protein (L)) and two non-structural proteins (C and V) [6]. PPRV is further characterized by one serotype and four different genetic lineages (LI to LIV) [7,8].

PPR is a fatal disease which causes high fever accompanied with a depression in general behavior. The disease is characterized by oculo-nasal discharges, erosive lesions of the nasal and oral mucous membranes, and respiratory signs in combination with gastrointestinal problems or abortions in female goats [9]. The pathogens that are considered relevant for the differential diagnostics of PPR are Foot-and-mouth disease virus (FMDV), Parapoxvirus ovis, Capripox viruses, and *Mycoplasma capricolum subsp. capripneumoniae* (Mccp) [6,10]. Mixed infections have been reported [11,12,13]. Foot-and-mouth disease (FMD) is a disease of worldwide concern that causes pyrexia as well as vesicular lesions of the nasal and oral mucosa and also on extremities and genital organs. Affected animals show excessive salivation and lameness [14,15]. Parapoxvirus ovis, also known as “Orf” or contagious ecthyma, is distributed worldwide. In young animals, Parapoxvirus ovis causes papules of the fascial mucous membranes, while in adults such lesions are also seen at the udder skin, in the inguinal area, and on the thigh [10]. Goatpox virus (GTPV) is mainly distributed across North and Central Africa and Asia [16]. Susceptible hosts are goats and sheep [17] that suffer from fever, inflammation of the facial mucous membranes, excessive salivation, and pox-like nodules disseminated over the hairless parts of the skin and mucous membranes [17,18]. Clinical signs also include dyspnea, depression, and reduced feed intake [18]. Mccp triggers a disease mainly focused on the respiratory tract of goats [19]. It is mainly present in countries in Africa, the Middle East, and Asia [20]. Clinically, animals suffer from high fever associated with lethargic behavior and a loss of appetite, as well as respiratory signs accompanied with nasal discharge, painful coughing, labored and accelerated respiration, and dyspnea [19,21].

PPRV was first identified in 1942 in Ivory Coast. Since then, the virus has spread rapidly towards East and North Africa and later also to Asia [22]. Currently, PPR occurs in over 70 countries, threatening 80 % of the sheep and goat population in the world. The extensive spread of this highly contagious disease shows its increasing relevance and its threat to animal husbandry [22,23]. With regard to the efficient control of this disease and the intended global eradication as laid out by the Food and Agriculture Organization of the United Nations (FAO) and the World Organization for animal health OIE, a fast and reliable diagnostic tool for testing suspicious animals is required [24].

In recent years, real-time reverse transcription recombinase polymerase amplification (RT-RPA) assays were developed for the simple and rapid molecular detection of PPRV in resource-limited settings. RT-RPAs deliver results within 20 min, thus being less time-consuming for a molecular diagnostic test, and also show a good diagnostic performance compared to RT-qPCR. However, these molecular assays have shown drawbacks regarding their detection limit and their reproducibility when compared to RT-qPCRs [25]. Another promising molecular detection tool that is considered to be suitable for point-of-care (POC) testing is the reverse transcription loop-mediated isothermal amplification (RT-LAMP) assay [26] due to its cost-effective, less sophisticated requirements and its detection times ranging from 20 to 55 min [27,28]. RT-LAMP assays were described for the detection of PPRV nucleic acid and have been demonstrated to meet the diagnostic sensitivities and specificities of RT-qPCR assays [27,28,29]. However, for RT-LAMP the design of primers is more complicated and thus more sophisticated.

Several rapid detection methods for PPRV have been developed and are partly ready to use [25,27,29,30,31]. In particular, lateral flow devices (LFDs) seem to be suited for a fast diagnosis in the field because they are easy to handle and to transport, available at low cost, and deliver results in less than 30 min [32,33].

However, while antigen-detecting LFDs are highly specific, they show deficiencies in sensitivity. Thus, negative samples that have been examined with an LFD detecting PPR antigens should be sent for laboratory confirmation [34,35]. Besides the rapid tests based on antigen detection, well-established RT-qPCR assays are available [36,37,38,39,40] and deliver reliable results within a few hours.

The present study aimed to develop a rapid test for PPRV that combines the requirements of a pen-side test based on rapid detection [26] with the advantages of a molecular detection method [41]. For this purpose, a combination of a markedly shortened extraction protocol and a high-speed RT-qPCR for PPRV was developed and tested. To facilitate storage and transport [42], a lyophilized master mix kit was used which offers the advantage of storage without a cold chain. The new fast-cycling approach was run on several qPCR cyclers for comparative evaluation. Finally, the protocol was extended for the detection of further pathogens with clinically similar signs similar to PPR to allow differential diagnostics. The validation of the test was performed with sample materials containing PPRV of all four lineages, FMDV, Parapoxvirus ovis, GTPV, and Mccp in single as well as in mixed samples. The new test system was designated as “FastCheck^FLI^ PPR-like”.

The idea behind the “FastCheck^FLI^ PPR-like” was to develop a diagnostic tool that focuses on fast and flexible PPRV diagnostics integrating the detection of additional pathogens, since the clinical similar signs of that pathogens complicate reliable diagnostics in the field. The different components are designed as a modular nucleic acid extraction and real-time RT-PCR detection system to achieve as much flexibility as possible. The presented data of our study should support the molecular diagnostics and differential diagnosis of PPR in small ruminants using universal available nucleic acid extraction and PCR applications.

## 2. Materials and Methods

The study design for the establishment of “FastCheck^FLI^ PPR-like” contains a couple of sole experiments that results in the final validation tests (Figure 1).

### 2.1. Rapid Nucleic Acid Extraction

A log 10 dilution series (10^−1^ to 10^−7^) was prepared with negative caprine saliva material and the PPRV isolate “Kurdistan/2011” (LIV) that contained a virus titer of 10^5^ fifty-percent tissue culture infective dose (TCID_50_)/mL. For nucleic acid extraction, a BioSprint 15 platform (Qiagen, Hilden, Germany) was used due to its simple and compact implementation. It is identical in construction to the KingFisher mL Purification System (ThermoFisher Scientific, Waltham, MA, USA). The BioSprint 15 platform is an open extraction workstation for processing a maximum of 15 samples per run. The extraction kits NucleoMag^®^ VET (Macherey-Nagel, Düren, Germany; hereinafter referred to as the VET kit), MagAttract^®^ 96 cador Pathogen Kit (Indical, Leipzig, Germany; hereinafter referred to as the CADOR kit), and MagMAX™ CORE Nucleic Acid Purification Kit (ThermoFisher Scientific; hereinafter referred to as the CORE kit), were tested comparatively in order to select the best chemical reagents. Depending on the specifications of the kits, two magnetic bead-based extraction protocols were used as initial nucleic acid isolation procedures (original protocols). For the VET and the CADOR kits, the original protocol lasted approximately 18 min, while the duration of the CORE kit was about 35 min (Appendix A). In order to design a speed-optimized extraction protocol, the software BindIt v3.3 (ThermoFisher Scientific) was used. In a gradual test approach, the duration of each extraction step was reduced to the acceptable minimum. For the subsequent RT-qPCR, the standard protocol was run on the Bio-Rad CFX96™ Real-Time PCR Detection System (BioRad Laboratories Inc., Hercules, CA, USA), while the master mix contained reagents of the AgPath-ID™ One-Step RT-PCR Reagents (Thermo FisherScientific) and the Polci-mix (Table 1).

### 2.2. High-Speed RT-qPCR

Two PPRV-specific primer-probe mixtures, one published [40] (here referred to as “Polci-mix”) and one in-house assay called “PPRV-mix 6” (Table 1), were evaluated comparatively for their suitability in a speed-optimized RT-qPCR approach. For the Polci-mix, primers and probes are located within the N gene, and for the PPRV-mix 6 within the H gene. Furthermore, the Polci-assay is based on an MGB probe, while the PPRV-mix 6 is based on a TaqMan probe. The log 10 dilution series (10^−1^ to 10^−7^) used for the establishment of the high-speed RT-qPCR was based on the RNA of a cell culture-adapted PPRV isolate designated as “Ivory Coast/89” (LI). Test series were performed by using the lyophilized master mix kit Takyon™ Dry No Rox One-Step RT Probe MasterMix kit (Eurogentec, Seraing, Belgium) and the qPCR cycler Bio-Rad CFX96. The starting point for the high-speed RT-qPCR was a protocol with a duration of 1 h 43 min, and a temperature–time profile consisting of reverse transcription at 48 °C for 10 min, PCR initial activation at 95 °C for 3 min, followed by 45 cycles of two further cycling steps comprising denaturation at 95 °C for 10 s, and a combined annealing plus extension at 60 °C for 60 s (standard protocol). This temperature–time profile was optimized by reducing all the PCR steps to the minimum time.

### 2.3. Device Test of Different qPCR Cyclers

The qPCR cyclers (Table 2) CFX96 Touch™ Real-Time PCR Detection System (BioRad Laboratories Inc., Hercules, USA), AriaMx Real-time PCR system (Agilent Technologies Inc., Santa Clara, USA), Magnetic Induction Cycler (MIC; Biozym Scientific GmbH, Hessisch Oldendorf, Germany), LightCycler^®^ 2.0 Instrument (Roche Molecular Systems Inc., Pleasanton, USA), and LightCycler^®^ 96 Real-Time PCR System (Roche Molecular Systems Inc., Pleasanton, USA) were evaluated comparatively in order to test their fast cycling features. A log 10 dilution series containing seven stages (10^−1^ to 10^−7^) of RNA extracted from a cell culture-adapted PPRV isolate was prepared in RNA-safe buffer 50 (RSB 50, [44]). The PPRV isolate used belongs to lineage 1 and is designated as “Ivory Coast/89”. For the preparation of the master mix, the lyophilized Takyon™ Dry No Rox One-Step RT Probe MasterMix kit was applied since it was the only available lyophilized kit at that time. As primer probe mixture, the Polci-mix was used (Table 1). The test series were performed on all five qPCR cyclers with a standard and a speed-optimized RT-qPCR protocol (short protocol 5) (for detailed cycling protocols see Section 2.2 and Section 3.2, respectively). Furthermore, an evaluation targeting several features of the qPCR cyclers (simple handling, variable in the number of samples, technical properties, dimensions, duration, and sensitivities) was carried out.

### 2.4. Validation of Three Lyophilized Kits

Log 10 dilution series (10^−1^ to 10^−7^) were separately prepared for five pathogens (PPRV, FMDV, Parapoxvirus ovis, GTPV and Mccp) in RSB 50 and initially validated using the AgPath-ID™ One-Step RT-PCR Reagents (Thermo FisherScientific, Waltham, USA) for the RNA viruses, and the QuantiTect Multiplex-PCR Kit NO ROX (Qiagen, Hilden, Germany) for the DNA pathogens. 

Subsequently, three lyophilized master mix kits were comparatively evaluated: the Takyon™ Dry No Rox One-Step RT Probe MasterMix kit (Eurogentec, Seraing, Belgium, hereinafter referred to as Takyon kit) can be stored at 15–35 °C for 18 months according to the manufacturer instructions, the Qscript lyo 1-step (Quantabio, Beverly, USA, hereinafter referred to as Qscript kit) is storable at room temperature for nine months, and the CAPITAL™ qRT-PCR Probe Mix (biotechrabbit GmbH, Heringsdorf, Germany, hereinafter referred to as Capital kit) is also storable at room temperature until the date of expiry (Table 3). All the kits were tested in five RT-qPCR approaches detecting PPRV, FMDV, Parapoxviruses, Capripoxvirues and Mccp in order to select the most suitable kit. The RT-qPCR master mixes were carried out in volumes of 12.5 and 20 µL, respectively. In the test series, the assays of all the target pathogens were carried out with both the standard protocol and the speed-optimized protocol for each lyophilized kit. The RT-qPCRs were carried out on the BioRad CFX96.

### 2.5. Validation of the “FastCheck^FLI^ PPR-Like” System

The validation panel consisted of ten samples containing one out of five target pathogens (single infection) and six samples containing more than two target pathogens (mixed infection). For this purpose, isolates of PPRV of all four lineages (Ivory Coast/89 (LI); Nigeria 75/1 (LII); Sudan/72 (LIII); Kurdistan/2011 (LIV); Indien/Shahjadpur (LIV); SMRV/UAE/2018/V135/Dubai (LIV)) as well as isolates of FMDV (A Iran 8/2015), Parapoxvirus ovis, GTPV (Indian GTPV) and Mccp (field sample from United Arabic Emirates, 2018) were prepared in negative caprine saliva. All the samples were examined in different combinations of extraction (original protocol vs. short protocol 4) and RT-qPCR (standard protocol vs. short protocol 5). For the validation test series, the NucleoMag^®^ VET extraction kit on the Biosprint 15 platform and the Qscript lyo 1-step for the RT-qPCR approach were used. The test series were performed on the BioRad CFX96 as reference device. Further tests were carried out on the mobile qPCR cycler Liberty16 (Ubiquitome Limited, Auckland, New Zealand), by which results can be analyzed with the corresponding Liberty16-App, version 1.7 (68), executed on an iPhone. In the final approach, eight wells per sample were needed: the first five wells for pathogen detection (PPRV, FMDV, Parapoxvirus ovis, GTPV, Mccp), two wells for internal control systems (EGFP-mix 1-FAM and β-Actin-DNA-mix 2-FAM), and one well as non-oligo-control (Table 1). In total, 16 samples imitating single and mixed infections were validated as described (Table 4 and Table 5).

## 3. Results

### 3.1. Speed-Optimized Rapid Extraction Protocols

For the establishment of a speed-optimized extraction protocol, the performances of three extraction kits were tested on the BioSprint 15 platform (Figure 2). In a step-by-step approach, the drying time of the beads could be reduced from four to one minute. Furthermore, several time-saving protocols were carried out during lysis/binding, all wash steps, and elution. Depending on the peculiarities of the extraction kits, speed-optimized extraction protocols for all three kits were established and the extraction protocol was shortened to approximately seven minutes. For the VET kit, the time reduction in short protocol 5 is possible down to a minimum of 6 min and 47 s (Appendix A). In short protocol 5, the mean deviation compared to the original protocol was more than one Cq-value (1.45; standard deviation SD 1.06), while for short protocol 4 the average deviation was less than one Cq-value (0.94; SD 0.68). Regarding the results of the CADOR kit, the extraction time could also be shortened to a minimum of 6 min and 47 s (Appendix A). For this kit, short extraction protocol 5 provided a dilution series with inconstant numerical values, meaning that the Cq-values between two dilution stages did not rise by around three Cq-values, as expected in a regular RNA dilution series and proven by the test results of the original extraction protocol (Appendix A). Better results were provided by short extraction protocol 4 with an extraction time of 7 min and 13 s (Figure 2, Appendix A). For short protocol 4, the mean deviation was also less than one Cq-value (0.81; SD 0.72) compared to the original protocol. For the CORE kit, all the protocols have been adapted to two washing steps according to the manufacturer´s specifications. Thus, the extraction time could be shortened to 6 min and 12 s (Appendix A) resulting in a dilution series with increasing numerical values (Appendix A) that showed Cq-steps of nearly the same amount between two consecutive dilution steps. The average deviation of short protocol 5 compared to the original protocol was more than two Cq-values (2.26; SD 0.72), while short protocol 4 showed a deviation of less than two Cq-values (1.75; SD 0.73). With short protocol 4, the first four out of seven dilution steps were found positive (Appendix A).

Comparing three different extraction kits, the VET kit and the CORE kit revealed the best results in the final speed-optimized extraction protocol compared to the original extraction protocol. For further validations, we decided to continue working with the VET kit due to the test results presented here. In summary, our final speed-optimized extraction protocol (short protocol 4) lasts approximately 7 min, enables the processing of up to 15 samples in parallel, and is operated on a portable, semi-automated extraction platform.

### 3.2. PPRV-Specific High-Speed RT-qPCR

The goal of this study part was a speed-optimized variant of the qPCR-protocol with still acceptable results in terms of analytical sensitivity. The applicability of two primer-probe mixtures (Polci-mix, PPRV-mix 6) was tested in a gradual shortening of the standard RT-qPCR protocol. Finally, a time-optimized PPRV-specific RT-qPCR of approximately 35 min (short protocol 5) could be established (Appendix A). The maximum shortened thermal profile consists of a reverse transcription at 48 °C for 1 min, initial activation at 95 °C for 2 min followed by 40 cycles comprising a denaturation step at 95 °C for 2 s, and a combined annealing plus extension step at 58 °C for 5 s (short protocol 5). 

The Polci-assay showed a reduction of 5 to 6 Cq-values in the dilution series of short protocol 5 (mean 5.53; SD 0.36) compared to the standard protocol. In addition, the short protocol showed a reduced analytical sensitivity of one log 10 dilution step (Figure 3). It also indicates a moderate decline in the numerical value of the relative fluorescence units (RFUs) compared to the standard protocol (Appendix A). In contrast, the rightward shift of the dilution curves was around three Cq-values for PPRV-mix 6 (mean 3.01; SD 0.18) along with a decreased analytical sensitivity of two log 10 dilution steps in the short protocol compared to the standard protocol (Figure 3). Interestingly, PPRV-mix 6 showed a greater drop in the RFU values than the Polci-assay (Appendix A). According to an explicit discrimination between negative and positive results, further validation tests were carried out with the Polci assay because of its stability in the RFU values and overall higher sensitivity.

### 3.3. Assessment of the Fast Cycling Features of Five qPCR Cyclers

Five qPCR cyclers were evaluated concerning several features that appear desirable for fast cycling and thus for a potential mobile detection unit for the point-of-care (POC) use (Table 2). The diagnostic performance of the qPCR cyclers was compared using the standard and the short protocol 5 (Figure 4). The best results in terms of the total run time were shown by the LightCycler 2.0, with 25 min in short protocol 5, but with a loss of up to three log 10 dilution steps in sensitivity (Appendix A). In addition, a mean deviation of 9.59 Cq-values (SD 1.54) was shown in the dilution series of short protocol 5 compared to the standard protocol. 

The AriaMx and the LightCycler 96 exhibited running times of 33 and 34 min, respectively. Concerning the increase in Cq-values, the AriaMx exhibited a reduction of 5 to 7 Cq-values in the dilution curves of short protocol 5 (mean 5.64, SD 0.68) compared to the standard protocol, but without any loss of sensitivity in the detection of the dilution steps in short protocol 5. For LightCycler 96, the deviation in Cq-values between both protocols varied between 2 to 6 (mean 4.18, SD 1.44). The reduction in analytical sensitivity was one log 10 dilution step in the short protocol compared to the standard protocol. The loss of sensitivity in short protocol 5 compared to the standard protocol as shown for both cyclers (Appendix A) should be acceptable based on the high viral load expected in clinically diseased animals. 

The running times of the qPCR cyclers CFX96 Touch and MIC were 38 and 39 min for the high-speed RT-qPCR protocol, respectively, accompanied with the loss of one log 10 dilution step. The results shown by the CFX96 Touch and MIC are also appropriate for the investigation of samples from acutely ill animals (Figure 4), as their shifts in Cq-values were 4 to 6 for the CFX96 Touch (mean 5.15, SD 0.34) and for the MIC (mean 4.75, SD 0.37) in the speed-optimized protocol compared to the standard protocol. 

For POC purposes, a simple handling of the software and a low weight according to frequent changes of the testing locations is recommended. The cyclers CFX96 Touch and AriaMx proved to be user-friendly in operation (integrated touch-screen function, intuitive operation, results are available directly on the device) (Table 2). The LightCycler 96 is also equipped with on-board-instrument diagnostics and touch screen options, but was less convincing regarding the self-explanatory and intuitive handling of the software. In contrast, the MIC is small in size and lightweight, but is not able to analyze and show the results directly on the device. For the analysis and interpretation of data, a personal computer is indispensable due to the small format of the qPCR cycler.

Overall, the AriaMx performed best according to the features compared here. Nevertheless, the BioRad CFX96 was used for the sole experiments of the study design due to its unlimited availability, while the qPCR cyclers of the device test were temporary borrowing equipment.

For validation purposes, another small-sized mobile qPCR cycler, Liberty16 (Ubiquitome, Auckland, New Zealand), was used due to technical preferences (battery operated, iPhone based), alleviating the need for extensive technical equipment (see Section 3.5.). The pen-side features were also collected for the Liberty16, but this mobile qPCR cycler was not part of our device assessment (Table 2) because of its recent market launch.

### 3.4. Evaluation of Lyophilized Kits for POC Testing

To facilitate the storage, transportation and use in the field, lyophilized chemicals are preferable because of their storage at room temperature and time-saving features during handling. Three different lyophilized kits were tested comparatively with the seven-fold dilution series of five pathogens (Figure 5). The Qscript kit is suitable for single samples because of its user-friendly format of 8-strip tubes. For the Takyon and Capital kits, the smallest available size after dissolution of the lyophilisate are 50 and 200 samples, respectively (Table 3). These kit sizes are less appropriate for the diagnosis of clinically diseased individual animals but well suited for epidemiological surveys or for diagnosing large animal husbandries. 

The standard RT-qPCR protocol compared to short protocol 5 was performed for the detection of PPRV, FMDV, Parapoxvirus ovis, GTPV and Mccp. The Qscript and Capital kits revealed similar results in terms of Cq-values in the standard and short protocol, thus being suitable for the speed-optimized RT-qPCR protocol when aiming to detect all five pathogens (Appendix A). The Takyon kit was less sensitive because of an increase in Cq-values and a drop in sensitivities for PPRV, Mccp, FMDV and Parapoxvirus ovis (Figure 5). These results plus practical aspects led to the further validation of “FastCheck^FLI^ PPR-like” with the Qscript kit.

### 3.5. Validation of the “FastCheck^FLI^ PPR-Like” Workflow

For the validation of “FastCheck^FLI^ PPR-like”, a panel of 16 samples was tested under laboratory conditions. Nucleic acid extraction was performed on the Biosprint 15 platform while a comprehensive test series of all the RT-qPCR protocols was carried out on the BioRad CFX96 to obtain reference data (Table 4). The portable qPCR cycler Liberty16 (Ubiquitome) was also integrated in the validation process (Table 5) because of its small-sized format and the possibility to interpret the results via an app on an iPhone. Besides, the Liberty16 can optionally be run as a battery-operated device, thus no permanent external electricity supply is needed, and the device seems to be well suited for POC testing.

The results of the test series carried out on the BioRad CFX96 showed a mean reduction in more than one of the Cq-values (mean shift 1.64; SD 1.91) when performing the “FastCheck^FLI^ PPR-like” in comparison to standard protocols. The impact of the speed-optimized RT-qPCR protocol is more pronounced than that of the shortened extraction protocol concerning the pathogen-specific RT-qPCR assays (Table 4). The internal controls, EGFP-mix 1 and β-Actin-DNA-mix 2, were detectable in all the samples, ensuring a reliable process control. Comparing these data with the results obtained by the Liberty16, the portable qPCR cycler produced slightly less sensitive results. The difference in the Cq-values of the identical test series performed on the BioRad CFX96 and the Liberty16 was more than one numerical values for both the standard (mean deviation 1.10; SD 1.37) and the short protocols (mean deviation 1.37; SD 2.09). Regarding the test series performed on the Liberty16, the combination of rapid extraction and high-speed RT-qPCR showed a mean reduction in more than one Cq-value (mean shift 1.72; SD 1.49) compared to the standard protocols. In two samples, the internal control system of the β-Actin-DNA-mix 2 failed to deliver positive results. Independent test series revealed that the cut-off for the Qscript kit has to be set on a Cq-value of 36 when applied on the Liberty16 (unpublished results), probably leading to the failure of the β-Actin-DNA-mix 2 (Table 5).

## 4. Discussion

The POC testing of transboundary and emerging diseases has become an important diagnostic advancement for a variety of animal pathogens [51,52,53,54,55]. In connection with disease control and eradication, critical decision-making close to suspected clinical cases is required [42,56]. Field-ready diagnostic methods should fulfil several requirements, such as detecting pathogens in a rapid and simple manner and delivering reliable results. In the best case, no further laboratory equipment should be required and the test should be ready-for-use in the field, which means that the device must be easy to handle and transport [26]. Numerous attempts have been made to develop devices that meet these requirements. LFDs, for instance, are small, light-weight devices capable of delivering results within several minutes. They are simple to use everywhere without much effort [32,52,53,55,57]. From the pathogens dealing in our study, only LFDs for the antigen detection of PPRV and FMDV are commercially available [30,31,52,55,58,59]. However, the diagnostic sensitivity of LFDs does not match that of the nucleic acid detection methods and is mostly reliable only for a limited range of sample matrices [30,33,35,56]. As shown for two commercially available LFDs detecting PPR antigen, the LFDs matched only a sensitivity between 53% to 75% compared to the RT-qPCR [35], but were cost-extensive as one test is commercially available for 8–11 Euros per test and antigen. The diagnostic sensitivity and specificity of the PPR-LFDs seems sufficient for the detection of PPRV antigen in strong PPR-diseased herds. Nevertheless, a negative result of the PPR-LFD can be caused by the insufficient sensitivity of the LFD in the individual test or by an infection with other pathogens, inducing PPR-like signs. Here, the “FastCheck^FLI^ PPR-like” strategy can be an option to expand the diagnostic possibilities. Further studies for the comparison of LFDs and the molecular “FastCheck^FLI^ PPR-like” using samples from animal trials or from the field will more clearly define the pros and cons of the different approaches. 

In contrast, standard nucleic acid detection methods are time-consuming, space-demanding, often expensive, need refrigeration units for the storage of reagents and rely on comprehensive sample preparation procedures. Thus, their use for POC testing is limited [60]. Hence, to overcome these disadvantages and to transfer these test systems into a valuable test format for POC testing [25,29], efforts were made to remodel RT-qPCR techniques for a quick, valid, and easy-to-use on-site application [44,51,54,61]. Mobile molecular test systems should also include rapid and semi-automated extraction procedures for the delivery of reliable results [60,62]. Thus, the present study aimed to combine both rapid extraction and high-speed RT-qPCR approaches for the development of a fast, reliable, and field-suited detection method for PPRV and clinically similar pathogens.

Besides the establishment of a variety of speed-optimized protocols for nucleic acid extraction and RT-qPCR, technological advances allowed the construction of compact, light-weight molecular detection devices that enable a versatile use and increased automation. Either portable thermocycler instruments [54,61] or portable molecular platforms that perform all steps from sample preparation to the result [56,62] are available.

Previously published data presented speed-optimized extraction protocols for different automated extraction platforms, whereof the BioSprint 15 reached the maximally shortened extraction time. Besides this, the BioSprint 15 revealed also a good correlation to manual extraction, but yielded a reduction in PCR efficiency. The mean PCR efficiency after extraction with the BioSprint 15 was 88.7 % compared to 101.2 % for manual extraction [63]. Due to its robust, easy-to-transport design, this open extraction platform seems to be a good choice for field investigations despite the decrease in PCR efficiency. The extraction kits used were based on a magnetic bead-based system because of its high extraction efficiency [60]. 

Large-sized, well-established qPCR cyclers (CFX 96 Touch, AriaMx, Light Cycler 2.0, Light Cycler 96) that were designed for routine laboratory use were tested and opposed to small-sized devices (MIC, Liberty16) that were developed for POC testing. The duration of an identical RT-qPCR protocol depends on the qPCR cycler used due to different ramping rates (Table 1), as previously reported [44]. 

Finally, a “FastCheck^FLI^ PPR-like” workflow was established that combines the speed-optimized extraction protocol and the high-speed RT-qPCR approach, aiming to transfer it to a field-ready format. The maximal shortened extraction protocol lasts approximately seven minutes, while the duration of a high-speed RT-qPCR protocol differs between the cyclers. The BioRad CFX96 needs 35 min and the Liberty16 needs 41 min for the established speed-optimized RT-qPCR protocol. The overall time frame of the “FastCheck^FLI^ PPR-like” workflow enables the investigation of several field samples in less than one hour. A shift of up to three Cq-values is considered acceptable under these time-optimized conditions. In fact, the results of our study indicate a less pronounced loss in sensitivity by rapid extraction than by high-speed RT-qPCR in the “FastCheck^FLI^ PPR-like” concerning the pathogen-specific RT-qPCR assays (Table 4). However, the mean deviation in Cq-values is around three (mean shift 3.01; SD 1.96) when comparing the “FastCheck^FLI^ PPR-like” workflow performed on the Liberty16 with the standard procedure carried out on the BioRad CFX96. Previously published data have shown marked differences in Cq-values between subclinically and acutely diseased animals [35]. Since, under field conditions, the diagnosis of acutely infected animals is mainly targeted, a high viral load of ocular and nasal swabs from diseased animals can be expected.

For the validation purposes of the here-developed “FastCheck^FLI^ PPR-like” workflow, a portable qPCR cycler—namely, Liberty16—and the qPCR cycler BioRad CFX96, a well-established qPCR cycler in centralized laboratories, were tested. The Liberty16 offers the advantage of being a small-sized (11 × 21 × 12 cm), portable, battery-operated, and iPhone-based device, thus no personal computer is required. Based on the format of the validated “FastCheck^FLI^ PPR-like” workflow and the technical conditions of the Liberty16, two field samples per run can be analyzed, including the detection of five additional pathogens, two internal controls, and a non-oligo-control. For the transfer of data, only a Bluetooth^®^ connection between an iPhone and the Liberty16 has to be established. The online transfer of data to an external e-mail account is feasible when internet is available.

The “FastCheck^FLI^ PPR-like” performed well on both qPCR cyclers. However, the results obtained by the Liberty16 were slightly less sensitive in comparison to the results of the BioRad CFX96 standard laboratory cycler. In comparison, the recently launched Liberty16 is specifically designed for POC diagnostics. Besides the Liberty16, further POC cyclers such as the Franklin™ (Biomeme Inc., Philadelphia, USA) and the MyGo Mini S (IT-IS Life Science Ltd., Dublin, Ireland) seem to be appropriate for POC test systems as well.

In comparison to further POC diagnostic tools, the new “FastCheck^FLI^ PPR-like” workflow delivers results in less than one hour, while results by LFDs take about 30 min [31,35]. Despite our improvements in “FastCheck^FLI^ PPR-like”, LFDs remain less dependent on additional equipment. However, a mobile detection unit can be implemented for the “FastCheck^FLI^ PPR-like” enabling the mobility of this molecular test system for field-investigations and including all necessary laboratory equipment (extraction platform, qPCR cycler, car battery, or power generator for power supply, extraction and PCR kits, pipettes and plastic goods e.g.) needed. The substantial pros of the “FastCheck^FLI^ PPR-like” workflow in contrast to the PPR-LFD are the investigation of several pathogens important in differential diagnostics [30].

As previously recommended, future perspectives for POC systems are the integration of controls in rapid molecular assays, as well as the use of smartphone-based detection units [64,65]. In the present study, it was demonstrated that “FastCheck^FLI^ PPR-like” meets these requirements and represents a highly versatile and secure approach for diagnostic purposes, in particular in endemic areas. Integrated internal control systems guarantee a secure and reliable diagnosis on the POC, while the iPhone based, small-sized qPCR cycler implements the idea of a portable diagnostic tool. 

The “FastCheck^FLI^ PPR-like” is designed as a modular test system consisting of various components (extraction platform, extraction kits, qPCR cyclers, lyophilized master-mix kits, RT-qPCR assays) that can be replaced by other devices, kits, or assays of further pathogens. However, the functionality of the replaced components has to be validated before application. Several well-established as well as recently launched kits and devices were compared in this study. “FastCheck^FLI^ PPR-like” delivers a versatile workflow, which is not limited to a certain pathogen. The modular design of “FastCheck^FLI^ PPR-like” enables the switch to manual extraction (e.g., column-based extraction) or to another molecular detection system (e.g., RT-LAMP or RT-RPA assays). Depending on the epidemiological situation, the extension of the new test system to additionally relevant pathogens producing clinical signs similar to PPRV is feasible. The principle in general may also be adapted to other clinical syndromes and the respective causative agents. In conclusion, “FastCheck^FLI^ PPR-like” was tested with a selected validation panel under laboratory conditions. In further studies, the diagnostic robustness of the “FastCheck^FLI^ PPR-like” must be proven under field conditions.

## Figures and Tables

**Figure 1 viruses-12-01227-f001:**
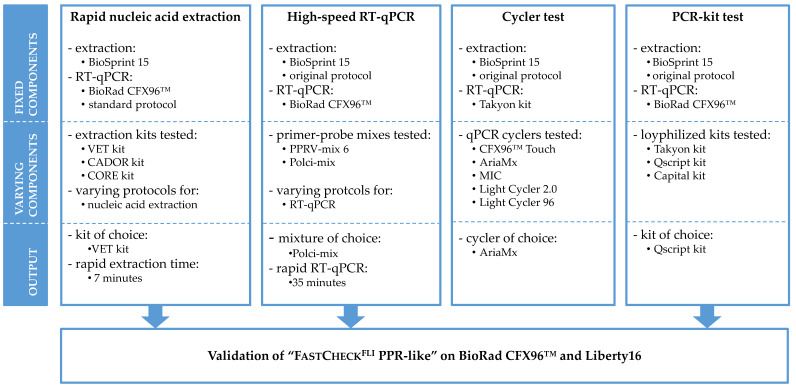
Workflow of the study design illustrating the sole experiments leading to the establishment of “FastCheck^FLI^ PPR-like”.

**Figure 2 viruses-12-01227-f002:**
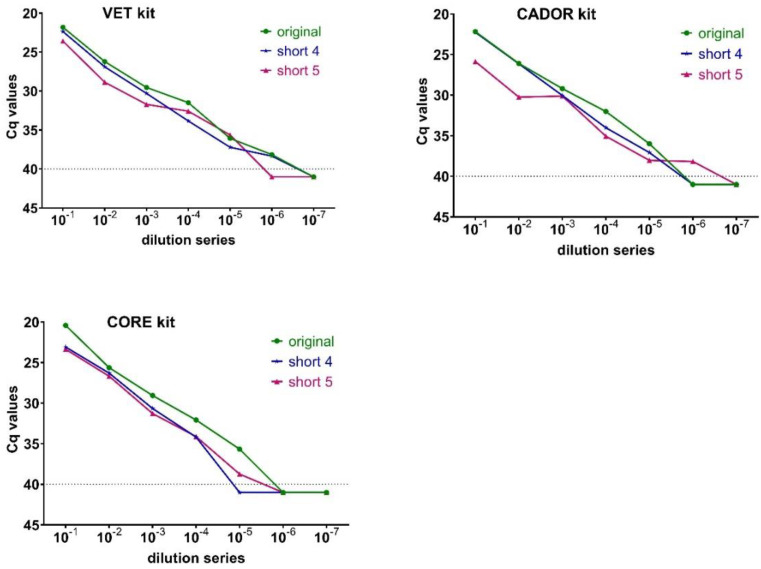
Test series for the establishment of a speed-optimized extraction protocol: differences in the Cq-values of the original (“original”) and the maximum shortened protocols (“short 4” and “short 5”) for three different extraction kits: VET kit; CADOR kit and CORE kit using a BioSprint 15 platform.

**Figure 3 viruses-12-01227-f003:**
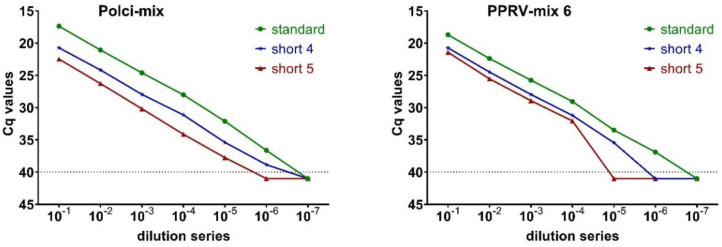
Test series for the establishment of a high-speed RT-qPCR: differences in the Cq-values of the standard (“standard”) and maximal shortened protocols (“short 4” and “short 5”) for two different primer–probe mixtures: Polci-mix and PPRV-mix 6.

**Figure 4 viruses-12-01227-f004:**
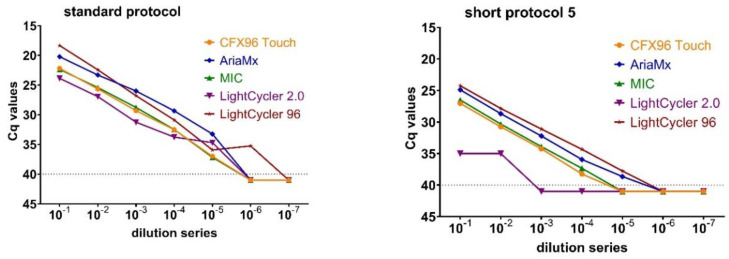
Device test with five qPCR cyclers: differences in Cq-values of the standard and short protocol 5 for the Polci-mix.

**Figure 5 viruses-12-01227-f005:**
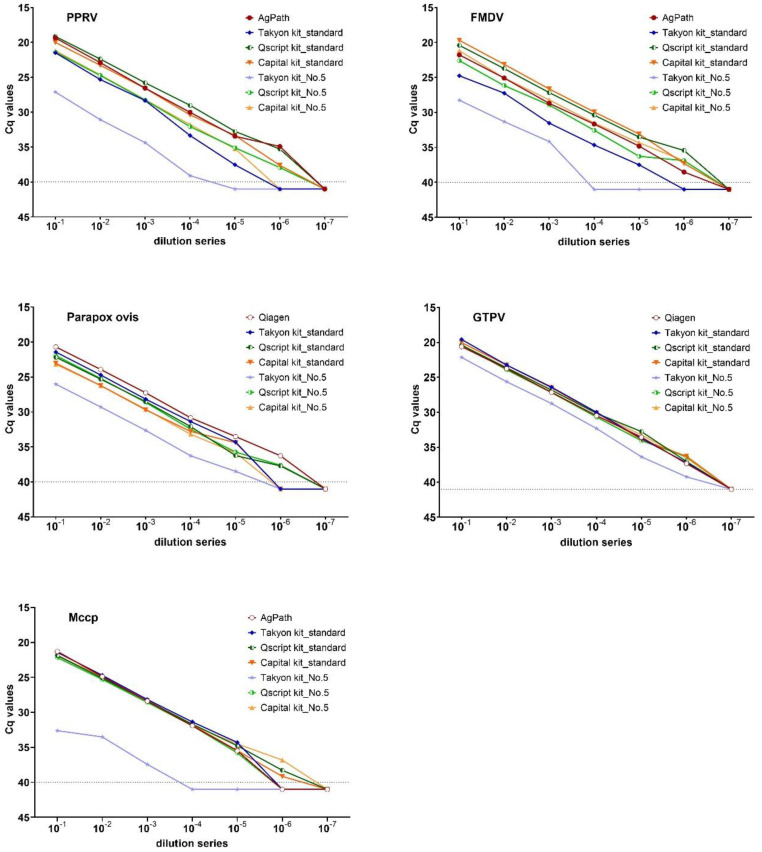
Test series with three lyophilized kits and five different pathogens: comparison of a well-established PCR protocol (named “AgPath” or “Qiagen”), the standard protocol (named “standard”) of the lyophilized kit, and the maximal shortened protocol (named “No.5”) of the corresponding kit. The kits used for the validation of the dilution series: AgPath = AgPath-ID™ One-Step RT-PCR Reagents; Qiagen = QuantiTect Multiplex-PCR Kit NO ROX. The three lyophilized kits used: Taykon kit; Qscript kit and Capital kit. The five different pathogens tested: PPRV, FMDV, Parapoxvirus ovis, GTPV, Mccp.

**Table 1 viruses-12-01227-t001:** Sequences of the primers and probes tested for a high-speed RT-qPCR approach for the detection of PPRV and clinically similar pathogens.

**PCR Assay**	**Genome Detection of**	**Primer/Probe**	**Sequence 5′–3′**	**Amplicon (Base Pair)**	**Reference**
Polci-mix	PPRV	PPR-Np-F298	CGC CTT GTT GAG GTA GTT CAA AGT	69	Polci et al., 2015 [40]
		PPR-Np-R366	ATC AGC ACC ACG TGA TGC A		
		PPR-NP-FAM-MGB	FAM– CAG TCC GGG TTG ACC T –MGBNFQ		
PPRV-mix 6	PPRV	PPR-H-8502-F	GAC CTC CYT CAT TTT GCA ATG G	85	in this study
		PPR-H-8586-R	ACT GAC YCT GAT CAC YCC GTA		
		PPR-H-8538-FAM	FAM-CCC RTG GTC AGA RGG GAG AAT CCC-BHQ1		
Righter	Mccp	Mccp-1F	CGC TCA CAT AGC CAA TCA TC	152	Righter et al., 2011 [43]
		Mccp-1R	TCG TTT TTA AGA GAA AAT CAA GCA		
		Mccp-1FAM	FAM-CAA GCT GAT GAA CAT AAA AAT GAT G-BHQ1		
IRES-3C	FMDV	FMD-IRES-3.1F	CTG GWG RCA GGC TAA GGA T	69	Wernike et al., 2013 [44] modified
		FMD-IRES-3R	CCC TTC TCA GAT YCC RAG TG		
		FMD-IRES-3FAM	FAM-CCC TTC AGG TAC CCC GAG GTA ACA-BHQ1		
Parapox-B2L	Parapoxvirus	PPV-B2L-455F	TCG ATG CGG TGC AGC AC	95	Nitsche et al., 2006 [45]
		PPV-B2L-539R	GCG GCG TAT TCT TCT CGG AC		
		PPV-B2L-FAM-MGB	FAM-TGC GGT AGA AGC C-MGB		
Capri-p32-mix1	Capripoxvirus	Capri-p32for	AAA ACG GTA TAT GGA ATA GAG TTG GAA	89	Bowden et al., 2008 [46] modified; Dietze et al. 2018 [47]
		Capri-p32rev	AAA TGA AAC CAA TGG ATG GGA TA		
		Capri-p32-FAM	FAM-ATG GAT GGC TCA TAG ATT TCC TGA T-BHQ1		
EGFP-mix 1	Enhanced green fluorescent protein gene	EGFP-1-F	GAC CAC TAC CAG CAG AAC AC	132	Hoffmann et al., 2006 [48]
		EGFP-2-R	GAA CTC CAG CAG GAC CAT G		
		EGFP-FAM	FAM-AGC ACC CAG TCC GCC CTG AGC A-BHQ1		
β-Actin-DNA-mix 2	beta-actin mRNA	ACT-1030-F	AGC GCA AGT ACT CCG TGT G	106	Toussaint et al., 2007 [49] modified; Wernike et al., 2011 [50]
		ACT-1135-R	CGG ACT CAT CGT ACT CCT GCT T		
		ACT-1081-FAM	FAM-TCG CTG TCC ACC TTC CAG CAG ATG T-BHQ1		

**Table 2 viruses-12-01227-t002:** Comparative evaluation of various qPCR-cyclers with regard to their usability as a pen-side test.

	CFX96 Touch	AriaMx	MIC	LightCycler 2.0	LightCycler 96	Liberty16
**Simple Handling of the Software**						
Intuitive	+++	++	++	-	+	++
On-board instrument diagnostics ^1^	Yes	Yes	No	No	Yes	No
Touch-screen option	Yes	Yes	No	No	Yes	Yes
**Cycler Equipment**						
Samples per instrument	96	96	48	32	96	16
Reaction vessels used	96-well plate	96-well plate	4 tube stripes	glass capillaries	96-well plate	8 strip PCR tube
Ramping rates (°C/s)	3.3–5.0	2.5–6.0	4.0–5.0	0.1–20	2.2–4.4	2.3
Heating (°C/s)	n.s.	6.0	5.0	n.s.	4.4	n.s.
Cooling (°C/s)	n.s.	2.5–3.0	4.0	n.s.	2.2	n.s.
Power supply	External	external	External	External	external	external and battery
**Dimensions**						
Width × deep × height (cm)	33 × 46 × 36	50 × 46 × 42	15 × 15 × 13	28 × 39 × 51	40 × 40 × 53	11 × 21 × 12
Weight (kg)	21	23	2.1	22	27	3.2
**Duration of a Single Run**						
Standard protocol	1 h 38 min	1 h 32 min	1 h 38 min	1 h 23 min	1 h 34 min	1 h 40 min
Short protocol 5	38 min	33 min	39 min	25 min	34 min	41 min

+++ = completely agree; ++ = rather agree; + = is insufficient; - = strongly disagree; ^1^ for analysis of the results, no personal computer is needed.

**Table 3 viruses-12-01227-t003:** Comparative evaluation of the three lyophilized master mix kits with regard to their usability at the pen side.

	Takyon Kit	Qscript Kit	Capital Kit
**Storage Conditions**			
Storage at … temperature	15–35 °C	room ~	room ~
Stability at room temperature for/until	18 months	9 months	expiry date
Storage after dissolution at … temperature	4 °C (for 24 h)	n.s.	−20 °C
**Features of the Kit (Manufacturer Specifications)**			
Reactions per kit (smallest size)	50	8	200
Recommended reaction size	20 µL	25 µL	20 µL
Smallest number of samples after dissolution	50	1	50
Delivery format of the lyophilizate	one tube	8-strip tubes	one tube

Takyon kit = Takyon™ Dry No Rox One-Step RT Probe MasterMix kit from Eurogentec; Qscript kit = Qscript lyo 1-step from Quantabio; Capital kit = CAPITAL™ qRT-PCR Probe Mix from biotechrabbit; n.s. = not specified.

**Table 4 viruses-12-01227-t004:** Validation of “FastCheck^FLI^ PPR-like” on the qPCR cycler BioRad CFX96 (results are shown in Cq-values or as “No Cq” for negative results).

**(A) Extraction: Original Protocol (17 min); RT-qPCR: Standard Protocol on BioRad CFX96 (1 h 38 min)**								
	**Pathogen Detection**					**Control Assays**		
	**PPRV**	**FMDV**	**Parapoxvirus**	**Capripoxvirus**	**Mccp**	**EGFP-1-FAM**	**β-Actin-DNA-2-FAM**	**Non-Oligo control**
**Single Infection**								
Ivory Coast/89 (LI)	20.3	No Cq	No Cq	No Cq	No Cq	28.1	31.7	No Cq
Nigeria 75/1 (LII)	23.8	No Cq	No Cq	No Cq	No Cq	27.7	34	No Cq
Sudan/72 (LIII)	24.1	No Cq	No Cq	No Cq	No Cq	27.4	34.9	No Cq
Kurdistan/2011 (LIV)	22.1	No Cq	No Cq	No Cq	No Cq	27.3	36	No Cq
Indien/Shahjadpur (LIV)	19.3	No Cq	No Cq	No Cq	No Cq	27.8	34.1	No Cq
SMRV/UAE/2018/V135/Dubai (LIV)	25.2	No Cq	No Cq	No Cq	No Cq	27.5	34.5	No Cq
FMDV (A Iran 8/2015)	No Cq	24.8	No Cq	No Cq	No Cq	26.7	36.1	No Cq
Parapoxvirus ovis	No Cq	No Cq	29	No Cq	No Cq	30.8	33.9	No Cq
GTPV (Indian)	No Cq	No Cq	No Cq	28	No Cq	26.3	37.2	No Cq
Mccp	No Cq	No Cq	No Cq	No Cq	20	26.5	28.3	No Cq
**Mixed Infection**								
PPRV * + Mccp	23	No Cq	No Cq	No Cq	31.1	27.1	32.4	No Cq
PPRV * + FMDV	24.9	29.1	No Cq	No Cq	No Cq	27.5	35.1	No Cq
PPRV * + GTPV	30.1	No Cq	No Cq	24.5	No Cq	26.8	34.2	No Cq
FMDV + Mccp	No Cq	29.6	No Cq	No Cq	23.2	27.5	31.2	No Cq
GTPV + Parapoxvirus ovis	No Cq	No Cq	31.5	25.3	No Cq	26.8	34.2	No Cq
PPRV * + Mccp + Parapoxvirus ovis	30.9	No Cq	30.5	No Cq	22.4	26.3	30.1	No Cq
**(B) Extraction: Short Protocol (7 min); RT-qPCR: Standard Protocol on BioRad CFX96 (1 h 38 min)**								
	**Pathogen Detection**					**Control Assays**		
	**PPRV**	**FMDV**	**Parapoxvirus**	**Capripoxvirus**	**Mccp**	**EGFP-1-FAM**	**β-Actin-DNA-2-FAM**	**Non-Oligo control**
**Single Infection**								
Ivory Coast/89 (LI)	21	No Cq	No Cq	No Cq	No Cq	25.5	30.5	No Cq
Nigeria 75/1 (LII)	24.4	No Cq	No Cq	No Cq	No Cq	26	34.8	No Cq
Sudan/72 (LIII)	24.5	No Cq	No Cq	No Cq	No Cq	25.7	34	No Cq
Kurdistan/2011 (LIV)	23	No Cq	No Cq	No Cq	No Cq	25.4	32.7	No Cq
Indien/Shahjadpur (LIV)	19.5	No Cq	No Cq	No Cq	No Cq	25.4	31.4	No Cq
SMRV/UAE/2018/V135/Dubai (LIV)	25.6	No Cq	No Cq	No Cq	No Cq	25.4	33.4	No Cq
**(C) Extraction: Original Protocol (17 min); RT-qPCR: Short Protocol on BioRad CFX96 (35 min)**								
**Single Infection**								
Ivory Coast/89 (LI)	21.8	No Cq	No Cq	No Cq	No Cq	26.3	31	No Cq
Nigeria 75/1 (LII)	26.1	No Cq	No Cq	No Cq	No Cq	25.3	34.3	No Cq
Sudan/72 (LIII)	27.1	No Cq	No Cq	No Cq	No Cq	25.2	35.6	No Cq
Kurdistan/2011 (LIV)	24.6	No Cq	No Cq	No Cq	No Cq	25.2	35.1	No Cq
Indien/Shahjadpur (LIV)	21.5	No Cq	No Cq	No Cq	No Cq	25.5	33	No Cq
SMRV/UAE/2018/V135/Dubai (LIV)	27.1	No Cq	No Cq	No Cq	No Cq	25.9	36.3	No Cq
**(D) Extraction: Short Protocol (7 min); RT-qPCR: Short Protocol on BioRad CFX96 (35 min)**								
	**Pathogen Detection**					**Control Assays**		
	**PPRV**	**FMDV**	**Parapoxvirus**	**Capripoxvirus**	**Mccp**	**EGFP-1-FAM**	**β-Actin-DNA-2-FAM**	**Non-Oligo control**
**Single Infection**								
Ivory Coast/89 (LI)	23.4	No Cq	No Cq	No Cq	No Cq	26.3	33	No Cq
Nigeria 75/1 (LII)	27.2	No Cq	No Cq	No Cq	No Cq	25.9	37.3	No Cq
Sudan/72 (LIII)	29	No Cq	No Cq	No Cq	No Cq	25.9	38.1	No Cq
Kurdistan/2011 (LIV)	26	No Cq	No Cq	No Cq	No Cq	25.9	36.2	No Cq
Indien/Shahjadpur (LIV)	22.8	No Cq	No Cq	No Cq	No Cq	25.8	36	No Cq
SMRV/UAE/2018/V135/Dubai (LIV)	28.1	No Cq	No Cq	No Cq	No Cq	26.1	38.5	No Cq
FMDV (A Iran 8/2015)	No Cq	27.5	No Cq	No Cq	No Cq	26.2	36.2	No Cq
Parapoxvirus ovis	No Cq	No Cq	31.4	No Cq	No Cq	26.4	35.6	No Cq
GTPV (Indian)	No Cq	No Cq	No Cq	29	No Cq	26.9	38.1	No Cq
Mccp	No Cq	No Cq	No Cq	No Cq	21.8	28.1	31.8	No Cq
**Mixed Infection**								
PPRV * + Mccp	26.3	No Cq	No Cq	No Cq	32.3	27.1	36.2	No Cq
PPRV * + FMDV	28.6	31.7	No Cq	No Cq	No Cq	27.2	38.2	No Cq
PPRV * + GTPV	34.7	No Cq	No Cq	26.5	No Cq	27.3	36.9	No Cq
FMDV + Mccp	No Cq	32.8	No Cq	No Cq	24.4	28.2	34.2	No Cq
GTPV + Parapoxvirus ovis	No Cq	No Cq	33.1	27.5	No Cq	27.1	37.4	No Cq
PPRV * + Mccp + Parapoxvirus ovis	34.5	No Cq	31.7	No Cq	24	27.6	33	No Cq

* PPRV isolate used for mixed samples: Kurdistan/2011 (LIV).

**Table 5 viruses-12-01227-t005:** Validation of “FastCheck^FLI^ PPR-like” on the qPCR-cycler Liberty16 (results are shown in Cq-values or as “No Cq” for negative results).

**(A) Extraction: Original Protocol (17 min); RT-qPCR: Standard Protocol on Liberty16 (1 h 40 min)**								
	**Pathogen Detection**					**Control Assays**		
	**PPRV**	**FMDV**	**Parapoxvirus**	**Capripoxvirus**	**Mccp**	**EGFP-1-FAM**	**β-Actin-DNA-2-FAM**	**Non-Oligo control**
**Single Infection**								
Ivory Coast/89 (LI)	22.6	No Cq	No Cq	No Cq	No Cq	29.4	30.3	No Cq
Nigeria 75/1 (LII)	26.1	No Cq	No Cq	No Cq	No Cq	29.1	34.6	No Cq
Sudan/72 (LIII)	25.5	No Cq	No Cq	No Cq	No Cq	28.8	34.5	No Cq
Kurdistan/2011 (LIV)	24.5	No Cq	No Cq	No Cq	No Cq	29.2	33.4	No Cq
Indien/Shahjadpur (LIV)	21.8	No Cq	No Cq	No Cq	No Cq	30.2	32.1	No Cq
SMRV/UAE/2018/V135/Dubai (LIV)	26.8	No Cq	No Cq	No Cq	No Cq	28.7	34.5	No Cq
FMDV (A Iran 8/2015)	No Cq	24.7	No Cq	No Cq	No Cq	29.2	34.1	No Cq
Parapoxvirus ovis	No Cq	No Cq	30.9	No Cq	No Cq	32.8	34.5	No Cq
GTPV (Indian)	No Cq	No Cq	No Cq	29.7	No Cq	28.5	36	No Cq
Mccp	No Cq	No Cq	No Cq	No Cq	22.8	29.3	29.2	No Cq
**Mixed Infection**								
PPRV * + Mccp	25.6	No Cq	No Cq	No Cq	32	28.6	34.4	No Cq
PPRV * + FMDV	26.5	29.8	No Cq	No Cq	No Cq	29.8	35.7	No Cq
PPRV * + GTPV	32.8	No Cq	No Cq	25.9	No Cq	28.8	34.5	No Cq
FMDV + Mccp	No Cq	29.2	No Cq	No Cq	21.9	29.6	30	No Cq
GTPV + Parapoxvirus ovis	No Cq	No Cq	32.5	26.9	No Cq	29	33.3	No Cq
PPRV * + Mccp + Parapoxvirus ovis	32.1	No Cq	32.2	No Cq	23.9	29.6	30.9	No Cq
**(B) Extraction: Short Protocol (7 min); RT-qPCR: Short Protocol on Liberty16 (41 min)**								
	**Pathogen Detection**					**Control Assays**		
	**PPRV**	**FMDV**	**Parapoxvirus**	**Capripoxvirus**	**Mccp**	**EGFP-1-FAM**	**β-Actin-DNA-2-FAM**	**Non-Oligo control**
**Single Infection**								
Ivory Coast/89 (LI)	25.8	No Cq	No Cq	No Cq	No Cq	29.7	34.8	No Cq
Nigeria 75/1 (LII)	30.3	No Cq	No Cq	No Cq	No Cq	29.4	35.2	No Cq
Sudan/72 (LIII)	30.4	No Cq	No Cq	No Cq	No Cq	27.9	36.5	No Cq
Kurdistan/2011 (LIV)	28.4	No Cq	No Cq	No Cq	No Cq	30	35.9	No Cq
Indien/Shahjadpur (LIV)	25.1	No Cq	No Cq	No Cq	No Cq	29.8	35.5	No Cq
SMRV/UAE/2018/V135/Dubai (LIV)	30.6	No Cq	No Cq	No Cq	No Cq	29.8	No Cq	No Cq
FMDV (A Iran 8/2015)	No Cq	26.9	No Cq	No Cq	No Cq	29.2	35.1	No Cq
Parapoxvirus ovis	No Cq	No Cq	32.6	No Cq	No Cq	32	34.9	No Cq
GTPV (Indian)	No Cq	No Cq	No Cq	31.1	No Cq	33.7	No Cq	No Cq
Mccp	No Cq	No Cq	No Cq	No Cq	22.5	33.5	32.4	No Cq
**Mixed Infection**								
PPRV * + Mccp	27.8	No Cq	No Cq	No Cq	32.2	29.4	36.7	No Cq
PPRV * + FMDV	29.2	30.9	No Cq	No Cq	No Cq	31.5	35.5	No Cq
PPRV * + GTPV	36	No Cq	No Cq	27.9	No Cq	31	35.7	No Cq
FMDV + Mccp	No Cq	31.8	No Cq	No Cq	25.5	29.9	33.7	No Cq
GTPV + Parapoxvirus ovis	No Cq	No Cq	33.6	28.8	No Cq	30.7	34.7	No Cq
PPRV * + Mccp + Parapoxvirus ovis	34.9	No Cq	32.5	No Cq	24.2	29.8	31.9	No Cq

* PPRV isolate used for mixed samples: Kurdistan/2011 (LIV).

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
