# Peer review of "“FastCheckFLI PPR-like”—A Molecular Tool for the Fast Genome Detection of PPRV and Differential Diagnostic Pathogens"

_viruses, 2020, doi:10.3390/v12111227_

Round 1

Reviewer 1 Report

Halecker et al. developed a fast method for the molecular diagnosis of PPRV and other pathogens that cause similar clinical signs. Several different conditions, reagents, primers, machines, and protocols were tested and compared. The amount of work is impressive and the manuscript is very well written. The developed method was properly tested and seems valid for its purpose. I am happy to recommend this manuscript for publication; however, there are few points that I think require some additional attention. First of all, the study is fairly complicated as it contains many elements and several experiments, and the manuscript is at times difficult to follow. I think that some efforts should be made in improving the clarity of some sections and the overall description of the procedure (see below for specific comments).

- It is not entirely clear what exactly FAST-CHECK is. Does it refer simply to the general idea of reducing isolation and PCR times or it also includes the use of the specifically selected kits and machines. At the beginning I thought that it included the methods described in section 2.5, but in the discussion you talk about the modularity of this method and the possibility of using different machines and protocols, including manual NA isolation. I think a better definition of the FAST-CHECK should be specified somewhere or at least you should give a recommendation on which combination of systems and protocols would be best to use.

- The study design is also at times unclear, given the huge number of experiments presented. I would suggest including a figure depicting the study design, the different parameters tested, and a sort of decision chart. Alternatively, a small introduction at the beginning of the M&M that describes which parameters were exactly testes and which material was used to test each step could be of help.

- Section 3.1. The overall result of what is the best system to use (or which one you recommend in this setting) should be added to this section. Furthermore, the Aria MX seems to be the one providing the most sensitive detection in one of the shortest times while having all the required features of low-weight and user-friendliness. Why was the CFX-96 used for the final validation?

- Figures 1-4. To facilitate reading these figures, that contain a lot of data, I suggest including some additional labelling to indicate what is what so the reader doesn’t have to refer to the caption and can more easily understand them. For example, in Figure 1 you can indicate “standard protocol” on top of the First figure, “short protocol 5” on top of the second, “Polci mix” before the first figure, and “PPRV-mix 6” before the third figure. In Figure 5, you can replace (A)-(E) with the pathogens’ names. Also, since there is no additional charge, you may consider using colors for the figures.

- Section 2.2. Which qPCR protocol/reagent/machines were used to test the different NA isolation procedures? Please specify.

- Section 2.3. What did you use to test the fast qPCR cycles? The same dilutions used in 2.1? Please, specify.

- Section 2.4. Which machine was used for these tests? Please, specify.

- Line 509. You mention here the word “cheap”. Indeed, I would not recommend the use of very expensive tests in these circumstances as they may reduce the testing capacity. What is the cost per test per-sample with your method and how does it compare to other POC tests (e.g. the LFD or LAMP)?

- Lines 573-5. I think this sentence should be accompanied by a big disclaimer since no actual comparison between the two system was performed and, as you also show, different PCR protocols/machine produce different results. This is also true especially considering that the BioSprint 15 used to extract NA, as you say at lines 518-25, has reduced isolation efficiency.

- Lines 592-4. I think that the limitation you mention here of the RT-LAMP can also apply to the FAST-CHECK, that also has a slight reduced sensitivity and was not designed in a multiplexed way. In fact, both the FAST-CHECK and LAMP can be performed in PCR strips with the same set-up you used in your study (5 samples, 2 internal controls, NTC). Furthermore, LAMP is probably cheaper and requires cheaper instruments. One disadvantage you can mention for the LAMP is that primer design is quite complicated and requires a more specialized expertise to develop the test.

Minor:

- Line 36: the L protein in morbilliviruses is the polymerase and it is therefore a non-structural protein.

- Lines 43-63: this section can be significantly shortened (or even entirely removed) since diagnosticsbased on clinical signs is not the main focus of this study. Such a long review of all the symptoms caused by these other pathogens makes the introduction much less fluid and the focus on the PPRV is lost. Given the diagnostic focus of this paper a review of currently used diagnostic methods (maybe moving in the intro some of the text from the discussion?) could be more informative than a long description of the disease caused by these pathogens.

- Line 50. Kids?

- Line 101. It would be of help if you could mention here that the speed-optimized protocol is referred to as “protocol 5”.

- Table one. What exactly does on-board instrument mean? Can an explanatory footnote be included here? Also, in the footnote there is a reference to “n. s. = not specified”, but I can’t see it anywhere in the table.

- Line 133. The reference to Table S3 belongs to the results section.

- Lines 138-42. This part should be moved to section 2.1, the first time these primers were mentioned.

- Line 166. You say here you used 12.5 and 20ul, but the table says 20 and 25.

- Line 286. What is the final speed-optimized protocol you picked, short4 or short 5?

- Table S3. The dash in the first column means that the -7 dilution wasn’t tested with the VET kit original length? Please specify.

- I would personally describe the optimization of the isolation method before comparing PCR machines as it would follow the order of procedural steps. However, this is just my personal opinion and feel free to ignore this comment!

Author Response

Reviewer 1

Halecker et al. developed a fast method for the molecular diagnosis of PPRV and other pathogens that cause similar clinical signs. Several different conditions, reagents, primers, machines, and protocols were tested and compared. The amount of work is impressive and the manuscript is very well written. The developed method was properly tested and seems valid for its purpose. I am happy to recommend this manuscript for publication; however, there are few points that I think require some additional attention. First of all, the study is fairly complicated as it contains many elements and several experiments, and the manuscript is at times difficult to follow. I think that some efforts should be made in improving the clarity of some sections and the overall description of the procedure (see below for specific comments).

#We appreciate your careful reading of the manuscript and the helpful comments you have shared. Your advices were well suited to structure the amount of data in this paper.

It is not entirely clear what exactly FAST-CHECK is. Does it refer simply to the general idea of reducing isolation and PCR times or it also includes the use of the specifically selected kits and machines. At the beginning I thought that it included the methods described in section 2.5, but in the discussion you talk about the modularity of this method and the possibility of using different machines and protocols, including manual NA isolation. I think a better definition of the FAST-CHECK should be specified somewhere or at least you should give a recommendation on which combination of systems and protocols would be best to use.

#The advice was noted and explanations were added (line 94-100, Figure 1).

The study design is also at times unclear, given the huge number of experiments presented. I would suggest including a figure depicting the study design, the different parameters tested, and a sort of decision chart. Alternatively, a small introduction at the beginning of the M&M that describes which parameters were exactly testes and which material was used to test each step could be of help.

#Thank you for reading the manuscript carefully. According to your suggestions a figure 1, illustrating the workflow of the studies, were added.

Section 3.1. The overall result of what is the best system to use (or which one you recommend in this setting) should be added to this section. Furthermore, the Aria MX seems to be the one providing the most sensitive detection in one of the shortest times while having all the required features of low-weight and user-friendliness. Why was the CFX-96 used for the final validation?

#Annotations were noted and a conclusion for section 3.1 was added. CFX96 was used as reference device due to its unlimited availability during the studies (line 321-323).

- Figures 1-4. To facilitate reading these figures, that contain a lot of data, I suggest including some additional labelling to indicate what is what so the reader doesn’t have to refer to the caption and can more easily understand them. For example, in Figure 1 you can indicate “standard protocol” on top of the First figure, “short protocol 5” on top of the second, “Polci mix” before the first figure, and “PPRV-mix 6” before the third figure. In Figure 5, you can replace (A)-(E) with the pathogens’ names. Also, since there is no additional charge, you may consider using colors for the figures.

#Your remarks were noted and the figures were changed accordingly.

- Section 2.2. Which qPCR protocol/reagent/machines were used to test the different NA isolation procedures? Please specify.

- Section 2.3. What did you use to test the fast qPCR cycles? The same dilutions used in 2.1? Please, specify.

- Section 2.4. Which machine was used for these tests? Please, specify.

#The advices were noted and remarks were added accordingly.

- Line 509. You mention here the word “cheap”. Indeed, I would not recommend the use of very expensive tests in these circumstances as they may reduce the testing capacity. What is the cost per test per-sample with your method and how does it compare to other POC tests (e.g. the LFD or LAMP)?

#Your remarks were noted and appropriate data were added when available.

- Lines 573-5. I think this sentence should be accompanied by a big disclaimer since no actual comparison between the two system was performed and, as you also show, different PCR protocols/machine produce different results. This is also true especially considering that the BioSprint 15 used to extract NA, as you say at lines 518-25, has reduced isolation efficiency.

#Your annotations were taken into account (Line 591-592).

- Lines 592-4. I think that the limitation you mention here of the RTLAMP can also apply to the FAST-CHECK, that also has a slight reduced sensitivity and was not designed in a multiplexed way. In fact, both the FAST-CHECK and LAMP can be performed in PCR strips with the same set-up you used in your study (5 samples, 2 internal controls, NTC). Furthermore, LAMP is probably cheaper and requires cheaper instruments. One disadvantage you can mention for the LAMP is that primer design is quite complicated and requires a more specialized expertise to develop the test.

#The annotations were noted and changed were made accordingly (Line 74-75).

Minor:

- Line 36: the L protein in morbilliviruses is the polymerase and it is therefore a non-structural protein.

#It is a great pleasure to get such a comprehensive and detailed feedback, thanks again for that. According to several references (Kumar et al., 2014; Munir, 2013) that are dealing with PPRV, C and V proteins are non-structural proteins while N, P, M, F, HN and L proteins are seen as structural proteins.

- Lines 43-63: this section can be significantly shortened (or even entirely removed) since diagnostic based on clinical signs is not the main focus of this study. Such a long review of all the symptoms caused by these other pathogens makes the introduction much less fluid and the focus on the PPRV is lost. Given the diagnostic focus of this paper a review of currently used diagnostic methods (maybe moving in the intro some of the text from the discussion?) could be more informative than a long description of the disease caused by these pathogens.

#Changes have been done accordingly.

- Line 50. Kids?

#That part of the introduction has been deleted.

- Line 101. It would be of help if you could mention here that the speed-optimized protocol is referred to as “protocol 5”.

#An additional remark was included.

- Table one. What exactly does on-board instrument mean? Can an explanatory footnote be included here? Also, in the footnote there is a reference to “n. s. = not specified”, but I can’t see it anywhere in the table.

#The annotations has been noted and changed accordingly.

- Line 133. The reference to Table S3 belongs to the results section.

#Table S3 has been cited later in the text.

- Lines 138-42. This part should be moved to section 2.1, the first time these primers were mentioned.

#Your annotations were noted. Due to the changes of the text order, the explanation according to these primers were maintained.

- Line 166. You say here you used 12.5 and 20ul, but the table says 20 and 25.

#I agree that the contrary information is a bit confusing. However, the volumes in the text and the table are equally correct.

- Line 286. What is the final speed-optimized protocol you picked, short4 or short 5?

#Your concern were noted and an appropriate remark added.

- Table S3. The dash in the first column means that the -7 dilution wasn’t tested with the VET kit original length? Please specify.

#Annotations were made accordingly.

- I would personally describe the optimization of the isolation method before comparing PCR machines as it would follow the order of procedural steps. However, this is just my personal opinion and feel free to ignore this comment!

#Your concern was shared and changes were adapted for a more comprehensible content of the manuscript.

References

Kumar, N., Maherchandani, S., Kashyap, S.K., Singh, S.V., Sharma, S., Chaubey, K.K. and Ly, H., 2014. Peste des petits ruminants virus infection of small ruminants: a comprehensive review. Viruses 6, 2287-327. doi:10.3390/v6062287

Munir, M. 2013. Peste des Petits Ruminants Virus. In: Munir, M. (Ed), Mononegavirales of Veterinary Importance, Vol. I: Pathobiology and Molecular Diagnosis, CABI, Wallingford, pp. 65-98

Reviewer 2 Report

The authors present in this publication adaptations of previously published methods, with the aim of reducing analysis time. To do this, the authors tested different thermal cyclers, different extraction kits and different RT-PCR amplification kits to reduce the time of the extraction protocols.

Minor modifications:

Line33: change Paramayxoviridae to paramyxovidiridae

Line 37: the cited publication is not the first  describing the 4 lineages of PPR, publications from 2010 and 2011 would be more appropriate

Line 141: the authors specify that “PPRV-mix6 within HN gene. " If this is the gene encoding for the hemagglutinin protein, then it should be identified as " H gene ".

Line143: the authors write "a protocol with duration of 1 hour 43 minutes ... (standard protocol)" but in table 1 the standard protocol for the CFX96 is only 1 hour 38 minutes. The authors should clarify this.

Major modifications

The comparison of the “FastCheckFLI” system with rapid diagnostic tests for PPR (LFD) is mentioned several times but none analysis has been carried out. The authors should perform this comparison in the laboratory based on the tests’ performances and also, to validate that using the system in field conditions in comparison with rapid PPR tests.

In order to reduce the time for an analysis, it is a good idea to reduce the time allocated to PCR amplification. Nevertheless, it would have been more adapted to initially choose the faster real-time PCR, at least faster than that published by polci et al, as it already exists in the literature. The authors did not explain why they chose the real-time PCR published by Polci et al, over one of those already published.

Comparing the amplification capacities of different real-time PCR machines is interesting, but does not provide very important information in this publication. The choice of a qPCR cycler by a laboratory is never decided taking into account only a particular method. This part of the manuscript should be reduced to the essentials.

The authors refer, repeatedly in the manuscript, the use of the "FastCheckFLI PPR-like" system uder field conditions. The need for two machines of more than 20Kg needing stable and continuous electric power, precision pipetting, protected from light, seem incompatible with this adaptation. Authors should review these arguments. This "FastCheckFLI PPR-like" system would be interesting for field use if the extraction and amplification systems were miniaturized. The system also requires ready-to-use amplification kits with primers and probes already integrated.

Author Response

Reviewer 2

The authors present in this publication adaptations of previously published methods, with the aim of reducing analysis time. To do this, the authors tested different thermal cyclers, different extraction kits and different RT-PCR amplification kits to reduce the time of the extraction protocols.

#We would like to thank the reviewer for the composition of this review. Your annotations and comments were noticed and text passages was changed in an according manner.

Minor modifications:

Line33: change Paramayxoviridae to paramyxovidiridae

#Your comment was noted and the spelling of the family name was corrected.

Line 37: the cited publication is not the first describing the 4 lineages of PPR, publications from 2010 and 2011 would be more appropriate

#Publications that date back to an earlier time were added.

Line 141: the authors specify that “PPRV-mix6 within HN gene. " If this is the gene encoding for the hemagglutinin protein, then it should be identified as " H gene ".

#We changed the abbreviation accordingly.

Line143: the authors write "a protocol with duration of 1 hour 43 minutes ... (standard protocol)" but in table 1 the standard protocol for the CFX96 is only 1 hour 38 minutes. The authors should clarify this.

#You are right that the durations are different. Please notice that the qPCR machines for both experiments, establishment of the high-speed RT-qPCR and device test, are different machines. In line 143 the BioRad CFX96 and for the device test the BioRad CFX96 Touch were used.

Major modifications

The comparison of the “FastCheckFLI” system with rapid diagnostic tests for PPR (LFD) is mentioned several times but none analysis has been carried out. The authors should perform this comparison in the laboratory based on the tests’ performances and also, to validate that using the system in field conditions in comparison with rapid PPR tests.

#The “FastCheckFLI PPR-like” platform is our idea of a rapid test for the differential diagnostic application for PPR-like diseased animals. In the present manuscript, preliminary work was done to establish methods and protocols for the simplification of the test procedure as a prerequisite for the evaluation of the modular platform in the field. We agree with your opinion that further tests, also in comparison to LFDs should be strived. According test are ongoing. Nevertheless, we believe and hope that the broad data presented in the manuscript can be helpful for other diagnosticians dealing with similar approaches.

In order to reduce the time for an analysis, it is a good idea to reduce the time allocated to PCR amplification. Nevertheless, it would have been more adapted to initially choose the faster realtime PCR, at least faster than that published by polci et al, as it already exists in the literature. The authors did not explain why they chose the real-time PCR published by Polci et al, over one of those already published.

#In preliminary studies, a comparison of different primer-probe-mixtures were performed and the Polci-mix delivered best results for the combination of a short extraction plus short RT-qPCR. To our knowledge, the turn-around-time depends on suitable primer-probe combinations, the temperature-time-profile, the cycler-features as well as on the PCR-kits used for amplification and detection.

Comparing the amplification capacities of different real-time PCR machines is interesting, but does not provide very important information in this publication. The choice of a qPCR cycler by a laboratory is never decided taking into account only a particular method. This part of the manuscript should be reduced to the essentials.

#The comments were well noted and the appropriate section was adjusted.

The authors refer, repeatedly in the manuscript, the use of the "FastCheckFLI PPR-like" system uder field conditions. The need for two machines of more than 20Kg needing stable and continuous electric power, precision pipetting, protected from light, seem incompatible with this adaptation. Authors should review these arguments. This "FastCheckFLI PPR-like" system would be interesting for field use if the extraction and amplification systems were miniaturized. The system also requires ready-to-use amplification kits with primers and probes already integrated.

#This is a misunderstanding. The extraction platform (BioSprint 15) as well as the RT-qPCR cycler (Liberty16) that are recommended for the “FastCheckFLI PPR-like” are chosen under the requirements to be light-weighting and portable. Besides, the Liberty16 is battery-operated, thus continuous electric power is not mandatory.

#However, we demonstrated preliminary data for a diagnostic tool that still have to be evaluated and validated under field conditions and thus, a mobile detection unit (comprising a car in the size of a van equipped with the necessary lab equipment needed) should be implemented. For this purpose further studies have to be strived.

Reviewer 3 Report

The manuscript of Halecker et al describes a number of investigations performed to develop a rapid means of detecting a number of pathogens which provide a differential diagnosis for PPR. Although I agree that the work is interesting (in terms of how fast we can push our existing extraction/PCR kits), I think that the vast amount of experimental work leads for a confusing paper overall. I am not sure that what has been described really support this new "molecular tool" as a single entity. The results section could be condensed to provide the most relevant data. Some specific queries are below: 

Line: 16 replace further with additional

Line 101: What were the cycling conditions for the standard protocols?

Line 131: This as written is confusing- please rewrite this sentence.

Line 135: Is this in table S1? if so, please refer to this in the text.

Line 141: Why is it important to state that the Polci probe is MGB? The Parapox assay is also MGB.

Line 153: The term "genome positive samples" is incorrect  in this instance since not every sample was detected. Log 10 dilution series... is sufficient.

Line 154: RSB 50 could be described.

Line 154: Initially validated is correct.

Line 166: Describe what an approach is? Is this the final volume of the reaction? How does this relate to the "five different approaches" mentioned in line 164.

Table 3: This table has a long title and could be amended. I would question the use of "various".

Line 183/184: How did protocol 4 and 5 evolve? Where were protocols 1-3? Why not call protocols 4 and 5 1 and 2, or A and B? 

Line 184: what is a test series?

Line 192: 16 samples seems a bit low- If this work is dependent on the assays designed by other, then reference to the specificity work done previously should be made.

Line 196: Was there any weighting given to these criteria? How were they evaluated?

Line 208 and elsewhere: I understand what is meant by rightward shift, but surely there is a better term that could be used?

Results section: this should be shortened as a whole.

Author Response

Reviewer 3

The manuscript of Halecker et al describes a number of investigations performed to develop a rapid means of detecting a number of pathogens which provide a differential diagnosis for PPR. Although I agree that the work is interesting (in terms of how fast we can push our existing extraction/PCR kits), I think that the vast amount of experimental work leads for a confusing paper overall. I am not sure that what has been described really support this new "molecular tool" as a single entity. The results section could be condensed to provide the most relevant data. Some specific queries are below:

#We appreciate the comments of the reviewer. Changes to the text has been conducted accordingly. We try our best to optimise the readability of the paper, also by a changed order of the manuscript and the introduction of a novel figure 1, summarizing the different parts of our work.

Line: 16 replace further with additional

#The advice was noted and corrections have been made accordingly.

Line 101: What were the cycling conditions for the standard protocols?

#Your remark was noted and cross-references were added during this section.

Line 131: This as written is confusing- please rewrite this sentence.

#The sentence was rewritten.

Line 135: Is this in table S1? if so, please refer to this in the text.

#The annotation was accepted and changes were made accordingly.

Line 141: Why is it important to state that the Polci probe is MGB? The Parapox assay is also MGB.

#In this context, the comparison only between PPRV-mix 6 and Polci-mix is essential because both primer mixes were compared regarding their RT-qPCR features during the shortening of the PCR-protocol. The Parapox-B2L assay was not part of this comparison. However, corresponding information for the PPRV-mix 6 was added.

Line 153: The term "genome positive samples" is incorrect in this instance since not every sample was detected. Log 10 dilution series... is sufficient.

#The remark was noted and corrections have been made accordingly.

Line 154: RSB 50 could be described.

#The abbreviation was already given in line 176.

Line 154: Initially validated is correct.

#The grammatically based error was corrected.

Line 166: Describe what an approach is? Is this the final volume of the reaction? How does this relate to the "five different approaches" mentioned in line 164.

#The annotation were noted and appropriate changes in the word choice were made.

Table 3: This table has a long title and could be amended. I would question the use of "various".

#Changes according to your remarks were made.

Line 183/184: How did protocol 4 and 5 evolve? Where were protocols 1-3? Why not call protocols 4 and 5 1 and 2, or A and B?

#The development of the short protocols are explained in the appropriate sections (section 2.1 and 2.2, respectively and section 3.1 and 3.2, respectively), thus additional explanations were renounced in this section. Besides, it was aimed to minimize the turn-around-time of the protocols (extraction and RT-qPCR, respectively) to shorten the duration for the “FASTCHECKFLI PPR-like”, thus only the most appropriate protocols (4 and 5, respectively) were used for validation purposes. Due to the amount of data, further protocols are available on request.

#The designation of the protocols is based on lab-internal terms in accordance with the development of further diagnostic methods. Thus, a renaming of the protocols would reduce the traceability of the raw data in a variety of validation tests.

Line 184: what is a test series?

#Perhaps the usage of the term “validation test” is more appropriate in this context. The word choice has been adjusted.

Line 192: 16 samples seems a bit low- If this work is dependent on the assays designed by other, then reference to the specificity work done previously should be made.

#You are right that a sample size of 16 seem to be a bit low for a comprehensive validation of a new test. Please note that several test series were carried out regarding the analytical sensitivity and specificity for all pathogen-specific assays (see supplemental material). Besides, the validation tests carried out in our studies should be seen as initial test series. As already mentioned in the discussion, further test series, preferable under field conditions, should be performed as proof of the diagnostic performance of the test.

Line 196: Was there any weighting given to these criteria? How were they evaluated?

#A weighting of the criteria was not strived.

Line 208 and elsewhere: I understand what is meant by rightward shift, but surely there is a better term that could be used?

#The remark was noted and the word “rightward shift” was replaced where possible.

Results section: this should be shortened as a whole.

#The section “Results” was edited. Please notice that the study design is very comprehensive, thus reduction of the data sets is restricted.

Round 2

Reviewer 1 Report

I am satisfied with the revisions. 

Reviewer 2 Report

  The answers and the modifications made help to make the article more coherent. I thank the authors for their efforts.

Reviewer 3 Report

Thank you for your responses to my queries.